# Meta-Analysis of the Association between Dietary Inflammatory Index (DII) and Colorectal Cancer

**DOI:** 10.3390/nu14081555

**Published:** 2022-04-08

**Authors:** Sharifah Saffinas Syed Soffian, Azmawati Mohammed Nawi, Rozita Hod, Mohd Hasni Ja’afar, Zaleha Md Isa, Huan-Keat Chan, Muhammad Radzi Abu Hassan

**Affiliations:** 1Department of Community Health, Faculty of Medicine, Universiti Kebangsaan Malaysia, Kuala Lumpur 56000, Malaysia; p108896@siswa.ukm.edu.my (S.S.S.S.); rozita.hod@ppukm.ukm.edu.my (R.H.); drmhasni@ppukm.ukm.edu.my (M.H.J.); zms@ppukm.ukm.edu.my (Z.M.I.); 2Clinical Research Center, Sultanah Bahiyah Hospital, Alor Setar 05400, Malaysia; huankeat123@yahoo.com (H.-K.C.); drradzi91@yahoo.co.uk (M.R.A.H.)

**Keywords:** Dietary Inflammatory Index, colorectal cancer, dietary pattern, modifiable risk factor

## Abstract

The Dietary Inflammatory Index (DII) was extensively used to examine the inflammatory potential of diet related to colorectal cancer (CRC). This meta-analysis aimed to update the evidence of the association between the DII and CRC across various culture-specific dietary patterns. Literature search was performed through online databases (*Scopus, Web of Science, PubMed*, and *EBSCOHost*). Observational studies exploring the association between the DII and CRC, published between 2017 and 2021, were included. The risk ratio (RR) and 95% confidence interval (CI) were separately computed for 12 studies comparing the highest and lowest DII scores and for 3 studies that presented continuous DII scores. A high DII score was associated with a higher risk of CRC (RR:1.16; 95% CI, 1.05–1.27). In the subgroup analysis, significant associations were seen in cohort design (RR: 1.24; 95% CI, 1.06–1.44), those lasting for 10 years or longer (RR: 2.95; 95% CI, 2.47–3.52), and in adjustment factor for physical activity (RR: 1.13; 95% CI, 1.07–1.20). An increase of one point in the DII score elevates the risk of CRC by 1.34 (95% CI: 1.15–1.55) times. The findings call for standardized measurement of the inflammatory potential of diet in future studies to enable the establishment of global guidelines for CRC prevention.

## 1. Introduction

Colorectal cancer (CRC) is currently the second leading cause of cancer-related mortality worldwide. In 2020 alone, approximately 1.9 million people were diagnosed with CRC, and 935,000 of them died within the same year [1]. The high burden of CRC in countries with a medium to high human development index (HDI) suggested the potential role of both sedentary lifestyles and dietary patterns in the development of CRC [2].

More than 60% of the overall CRC cases were reported as sporadic, occurring in people without genetic predisposition or family history of CRC [3]. Such a trend further points to the impact of modifiable risk factors in the CRC development. It is generally believed that long-term exposure to an unhealthy diet, physical inactivity, smoking, and alcohol consumption are all likely to trigger the chronic systemic inflammation, which eventually induces the proliferation of cancer cells [4,5]. Molecular studies also demonstrated that the production of pro-inflammatory cytokines and arrays of free radicals present at cellular levels were attributable to the consumption of an unhealthy diet [6,7,8].

Furthermore, numerous studies provided insight into the relationship between specific food items and CRC. For example, a high intake of red meats combined with a low intake of vegetables was shown to elevate the risk of CRC by 2.6 times [9]. In contrast, a high intake of fruits, cereals, nuts, and milk and dairy products lower the risk of CRC by 64% [10]. Another study also reported that the risk of CRC increased by 17% and 18%, respectively, as a result of consuming 100 g of red meats and 50 g of processed meat products daily [11]. Additionally, a recent study demonstrated that taking more than two servings of sugar-sweetened beverages per day heightened the risk of CRC by 16% [12].

There were various types of dietary assessment methods frequently used for epidemiological purposes, which includes the 24 h dietary recall, the dietary record, and the food frequency questionnaire (FFQ). However, they were unable to relate the specific dietary risk factor with levels of inflammatory markers well-established in CRC. The advent of the Dietary Inflammatory Index (DII) provides a quantitative means to study the relationship between the pro-inflammatory diet and CRC. It allows the assessment of the inflammatory potential of individual food items using an FFQ, by which a DII score can be calculated. A higher DII score suggests a stronger inflammatory potential of a food item [13]. Remarkably, DII has been utilized in substantial epidemiologic studies that include different ethnicities and various health outcomes. To date, the evidence on the usefulness of the DII in predicting the risk of CRC is established mainly focus on the Western diet with little input from other parts of the world based on the studies published before 2017 [14,15]. As CRC emerges as a major cancer type globally, this review was designed to update the evidence of the association between the DII and CRC based on the latest studies on a wide range of culture-specific dietary patterns.

## 2. Materials and Methods

The review was conducted according to the Preferred Reporting Items for Systematic Reviews and Meta-Analyses (PRISMA) guidelines to ensure transparency [16]. Guided by the PRISMA, the authors performed the systematic literature search through formulation of related research questions. The systematic searching process consists of identification, screening, and eligibility stage, which were performed for each database. The authors independently appraised the quality of included studies using the Newcastle–Ottawa quality assessment scale. Following that, the authors read through all full-text articles for data extraction and analysis.

### 2.1. Formulation of the Research Question

The research question was formulated based on the PICO concept: a tool often used to assist authors in developing suitable research questions for systematic review. It consists of population or problem, interest, and context or outcome. Based on this concept, the authors have included the three main aspects in the review: adults (population), Dietary Inflammatory Index (interest), and CRC (outcome), which led the authors to the main research question, “How does the Dietary Inflammatory Index determine the risk of CRC in various adult populations?”.

### 2.2. Systematic Searching Strategies

The systematic searching strategy was initiated with the identification, followed by screening and eligibility process (Figure 1).

### 2.3. Identification

A comprehensive literature search performed between 1 and 4 November 2021 through *Scopus, Web of Science, PubMed*, and *EBSCOHost*. These four online databases covered scientific publications in more than 30,000 journals [17,18]. The discoverability of articles was enhanced using synonyms of keywords and the medical subject headings, combined by using the Boolean operators (Table 1). A total of 5569 records were retrieved from the databases, and 542 duplicates were removed. The remaining records were then exported to a Microsoft Excel spreadsheet for screening.

### 2.4. Screening

The titles and abstracts of all the records were screened for eligibility. An article was retained for the analysis only if (i) it was published between 2017 and 2021, (ii) a full article was available, (iii) it described an observational study exploring the association between the DII and CRC, and (iv) it was published in English. The duration of published articles screened was determined based on the recent development of DII in cancer. Despite the consistency of DII as an instrument in substantial evidence-based research, the additional food parameters and revised scoring algorithms enhanced the comparability between studies [19]. Considering the utilization of revised version DII in recently published studies concerning colorectal cancer, the selection of the latest five years duration is justified. Review articles, editorials, proceedings, commentary articles, and articles focusing on cancers other than CRC were excluded.

### 2.5. Eligibility

Of the 118 full-text articles screened for eligibility, 103 were excluded. They ranged from animal studies (*n* = 38), clinical trials (*n* = 8), genetic studies (*n* = 13), molecular studies (*n* = 21), and studies on diseases other than CRC (*n* = 11) to studies on early-onset CRC (*n* = 6). The remaining 15 articles (9 case-control and 6 cohort studies) were subjected to the quality appraisal.

### 2.6. Quality Appraisal

The quality of the selected studies was assessed using the Newcastle–Ottawa quality assessment scale [20]. This scale was designed for nonrandomized studies, focusing on the sample selection, the comparability of study groups, and the ascertainment of exposure or outcome of interest. The total number of stars rewarded to a study indicated its quality, either low (≤3 stars), moderate (4–6 stars) or high (≥7 stars). The quality assessment of the selected studies was independently performed by two authors. Any disagreements between them were resolved by consensus, and a third reviewer was consulted when necessary. The results for the quality assessment are presented in Table 2. The quality of the studies ranged from moderate to high, and all of them were included in the meta-analysis.

### 2.7. Data Abstraction and Analysis

The information extracted from each included study ranged from the author’s names, year of publication, study location, study design, study period, study instrument used, sample size, type of data (categorical or continuous), measure of association (odds ratio (OR) or hazard ratio (HR) with the corresponding 95% confidence interval (CI)), range of DII scores, and number of food parameters to adjusted factors (Table 3). The random-effect meta-analysis was performed separately for 12 studies that compared the two categories with the highest (pro-inflammatory) and lowest (anti-inflammatory) DII scores and three studies that presented continuous DII scores. The Review Manager 5.4 (RevMan) software was used to generate the summary risk ratios (RRs) and the corresponding 95% Cis, while I^2^ statistics was used to test the heterogeneity of the studies. Given the heterogeneity of the studies that categorized the DII scores, a series of stratified analyses was also conducted based on the study design, region, study period, and adjusted factors (family history of CRC, educational level, comorbidities, physical activity, and BMI).

## 3. Results

The 15 studies included in the analysis covered ten countries from different regions, including the United States (*n* = 5) [22,28,30,32,33], the Netherlands (*n* = 2) [29,34], Argentina (*n* = 1) [14], Spain (*n* = 1) [24], Germany (*n* = 1) [31], Canada (*n* = 1) [26], Iran (*n* = 1) [25], Jordan (*n* = 1) [27], Korea (*n* = 1) [23], and China (*n* = 1) [21]. Geographically, most of the studies were from the regions of America (AMR) (*n* = 7) [14,22,26,28,30,32,33], followed by the European region (EUR) (*n* = 7) [24,29,31,34], the western Pacific region (WPR) (*n* = 2) [21,23] and the eastern Mediterranean region (EMR) (*n* = 3) [25,27]. The study period ranged from five years and less (*n* = 6) [23,24,25,26,27,31], six to ten years (*n* = 6) [14,21,28,29,30,34], and more than ten years (*n* = 3) [22,32,33].

The studies assessed 18 to 35 food parameters using the FFQ. The highest and lowest measures of association were observed in Iran (OR: 2.64; 95% CI: 1.40, 4.99) [25] and the United States (HR: 0.72; 95% CI: 0.46, 1.12) [33], respectively.

### Association between DII and the Risk of CRC

Despite the heterogeneity of the 12 studies that categorized DII scores (I^2^ = 69%, *p* = 0.0002), the pooled analysis on 111,702 individuals demonstrated a significant association between a high DII score and the risk of CRC (RR: 1.16; 95% CI, 1.05–1.27) (Figure 2).

Another analysis on 1175 individuals from three studies also showed that an increase of one point in the DII score elevated the risk of CRC by 1.34 (95%CI: 1.15, 1.55) times. Data synthesis stratified by dichotomous outcome for DII score (Figure 3) showed that high DII score group had increased 34% risk of CRC (RR:1.34; 95% CI, 1.15–1.55) in a random effect model with no heterogeneity (I^2^ = 0%, *p* = 0.000).

Due to the high heterogeneity result, several subgroup analyses on study design, groups for DII score, study settings based on WHO region, duration of study, and adjustment factors were conducted (Table 4). Stratified by the study design, positive association was found between DII score and risk for CRC in case control studies (RR: 1.14; 95% CI, 0.89–1.45) and cohort studies (RR: 1.24; 95% CI, 1.06–1.44) but with a high degree of heterogeneity. The summary RR indicated no heterogeneity for grouping of DII score either analysed in form of continuous data or categorical baseline. Notwithstanding that, the categorical form of data analysis presented significant increased risk for CRC by 61% (RR: 1.61; 95% CI, 1.26–2.05), much higher than that of continuous presentation of DII score (RR: 0.35; 95% CI, 0.28–0.41). Low heterogeneity (22%) was observed in studies conducted among the eastern Mediterranean region, which could possibly be due to sharing of culture with similar dietary pattern. In regions such as the Europe and Asia, high heterogeneity observed with 98% and 95% each, highlighted the influence of cultural blend within the study population that masked the dietary pattern. Subgroup analysis based on the study duration highlighted significant association with CRC risk when conducted for 10 years or more (RR: 2.95; 95% CI, 2.47–3.52). Positive relationship was seen in all studies with adjustment factors for family history of CRC, education level, comorbidities, physical activity, and BMI but with moderate to high heterogeneity. Low heterogeneity (55%) was seen in studies without adjustment for family history of CRC with RR: 1.31; 95% CI 1.10–1.56, and no heterogeneity in studies without adjustment for physical activity with RR: 1.13; 95% CI 1.07–1.20.

## 4. Discussion

This meta-analysis confirmed the association between high DII scores and the increased risk of CRC. It was grounded on the standardized quantification of inflammatory markers produced individual food parameters, which has long been used to predict the risk of chronic non-communicable diseases including cancer [35,36]. Instead of giving general recommendations on high-fat and low-fiber food intake, the DII enables an objective assessment of dietary patterns and the risk of cancer at an individual level [21,37,38]. While the Asian population contributed to more than half of the CRC cases [2,39], this study fills the gap by providing evidence on a wider range of culture-specific dietary patterns across the world based on the latest studies.

In the subgroup analysis, cohort studies showed a more significant association between high DII score and increased risk for CRC, as compared to case-control studies. As cohort studies allow a better control of confounding factors in nature, such finding implies a stronger impact of pro-inflammatory dietary patterns on the CRC development as compared with the study duration and other adjustment factors. It is also worth highlighting that a relatively low degree of heterogeneity in the studies from the EMR supported the notion that the Mediterranean diet had protective effect against risk for CRC, reflecting the role of specific dietary pattern that is possibly culturally driven.

Additionally, when presented in categorical form, DII score demonstrated significant increased risk for CRC rather than interpreted in continuous per-one-unit increment. The most commonly used classification in the selected studies was based on three categories and hence provides easy and convenient comparison. Nevertheless, future validation studies on standardization of DII score calculation are necessary to strengthen the interpretation against risk for CRC and act as a baseline for guideline development. Even though adjustment factors studied showed proportionate relationship with CRC risk, the high degree of heterogeneity obtained should be considered cautiously.

Although the pooled RR for CRC in the study was lower than that published by existing literatures, the findings corroborated the positive association between high DII score and increased risk for CRC as evidence in other studies. In a review comparing four studies among the Western population, more proinflammatory diet scores were linked with a 12–65% higher risk of CRC compared to the anti-inflammatory diets [40]. Similarly, an increase of one point in the DII score was suggestive of elevating the risk for CRC by 7% in a relatively more heterogenous population [41]. The range of risk difference indicate the sensitivity of culture-specific dietary patterns as incorporated in the current meta-analysis. While the Westernized diet has been frequently examined to exert pro-inflammatory effects that further enhances the risk for CRC, limited studies are available to support the role of dietary pattern in the non-Western population. Thus, more studies are warranted to explore the culture-specific dietary pattern particularly in the settings of diverse, multiracial Asian countries.

The most common approach used to measure the adherence to a routine healthy dietary intake is the Healthy Eating Index (HEI) or the Mediterranean diet since the indices were based on certain dietary recommendations [32]. Overall, the Mediterranean diet composed of rich, plant-based foods, including the fruits, vegetables, nuts, legumes, and whole grain products, combined with regular intake of seafood and low intake of red and processed meat [42]. Although many studies relate the practice of Mediterranean diet with less risk for CRC, in regions where plant-based foods serve as the food staple, the incidence of CRC continues to rise [43]. Future research works should consider the influencing factors concomitant to dietary pattern that could possibly contributed to the progression of CRC over time.

High intake of processed and red meat has been linked to increased risk of CRC. However, in countries like Korea where the CRC incidence rate is relatively high, with 44.5 cases per 100,000 persons per year [44], studies reported less than 20% of the population consumed processed meat products [45], indicating the presence of other factors that may influence the tumor progression. Numerous empirical studies proposed that the presence of mutagenic compounds, such as the heterocyclic amines (HCAs), polycyclic aromatic hydrocarbons (PAHs), and acrylamides formed during food preparation, raised the risk of CRC [24,36,46]. Understanding the cooking method and food processing involved provide beneficial value to explain the impact of food culture in the complex mechanism of CRC.

The intake of dietary fiber on daily basis has crucial role in the functionality of the guts to enhance transit time and stool formation. Disturbance of the physiologic system affected by the food solubility and fermentability causes tremendous effects towards the gut lining, thus potentially causing tumorigenesis. Notwithstanding that, a recently published longitudinal study revealed that consumption of dietary fiber intake improved the physical function and overall health of the CRC survivors upon completion of their treatment [47]. Changing to healthier diet pattern containing more anti-inflammatory foods had protective effect against cancer recurrence as well as prolonged survival [9,15,35,38]. Therefore, extensive health campaigns and awareness towards a healthy, balanced diet should target the average-risk groups and be initiated as early as possible.

Other important confounders considered in the studies included age and sex. When comparing the DII score across opposite gender, few studies suggested sex differentials pertaining to the risk of CRC [14,24]. A comparison study on Canadian populations showed that a 33% reduction in the risk of CRC in men [41] was attributable to the intake of anti-inflammatory diets, whereas no significant association was observed with similar dietary pattern among women. More studies are required to explain the role of sex differentials and dietary intake against the risk for CRC. Similarly, when accounted together with the physical activity factor, high DII score showed tendency to raise the CRC risk among individuals with sedentary lifestyle [22]. Collectively, this explained the importance of healthy lifestyle of an individual in a holistic manner whereby the modifiable risk factors are interdependent, leading to rapid progression towards CRC [48].

The interplay of modifiable risk factors, such as unhealthy dietary patterns and sedentary lifestyles, have partly contributed to the complexity of obesity to an extent [49]. Several epidemiological evidence have demonstrated that individuals with higher-than-normal BMI are likely to have a higher risk for CRC [8,49,50,51,52]. In a large, population-based cohort trial, obesity during early adulthood and a constantly increasing BMI throughout the lifespan were significantly associated with CRC [53]. Recent genetic studies suggested that for every one-unit increase in genetically predicted BMI, there is an increase in the odds ratios for CRC [54,55,56], indirectly implying the causality relationship between obesity and CRC. Although the current review reported statistically significant heterogeneity between studies with adjustment for BMI, future studies should consider the influence of residual confounding to delineate the true effect of BMI on CRC.

The review provides evidence on potential useful instrument for standardized comparison regarding the inflammatory potential of diet pertaining to CRC using an indexing approach. The score of DII can be interpreted both in continuous and categorical groups, contributing to the dynamic of its usability in the interpretation of future cancer-related research. Provided that the Asian region has the highest burden of CRC worldwide and that dietary intake is culturally specific, studies that explore the nutritional risk factors among multiracial Asian population are warranted. Moreover, the standardized calculation method of DII across all studies increased the comparability and thus also provides a valid description.

The limitation of the review includes the heterogenous nature of studies, including the study population characteristics, sample size, study design, and follow-up periods. The questionnaires used for food assessment were different and thus contributed to the recall bias. Nutritional assessment that excludes specific culture food items limits the ability to explore further regarding local influence on dietary pattern. Furthermore, the baseline calculation of DII score considered in the studies may not represent the true long-term dietary pattern, as adult diets vary over time. Despite substantial heterogeneity observed across the studies, subgroup analyses were performed to explore the source of heterogeneity. Grouping of DII score, study settings and adjustment factor for physical activity were likely related to the heterogeneity trend. Nonetheless, inevitable small sample size and confounding factors in each study contributed to the limitation potentially affecting the result.

## 5. Conclusions

Substantial evidence supported the association between inflammatory potential of diet and the development of CRC. However, future research involving multiethnic population, such as in Asian regions, is needed to explain the climbing CRC incidence and plan for preventive intervention strategies. Concerted efforts tailored to specific-culture dietary patterns call for multisectoral engagement in program planning to ensure effective outcome. Quantitative evidence as reported by the DII score and the impact towards CRC risk in the review helps to advocate to the public health authorities about the importance of tackling the underlying factors that shape the dietary pattern.

## Figures and Tables

**Figure 1 nutrients-14-01555-f001:**
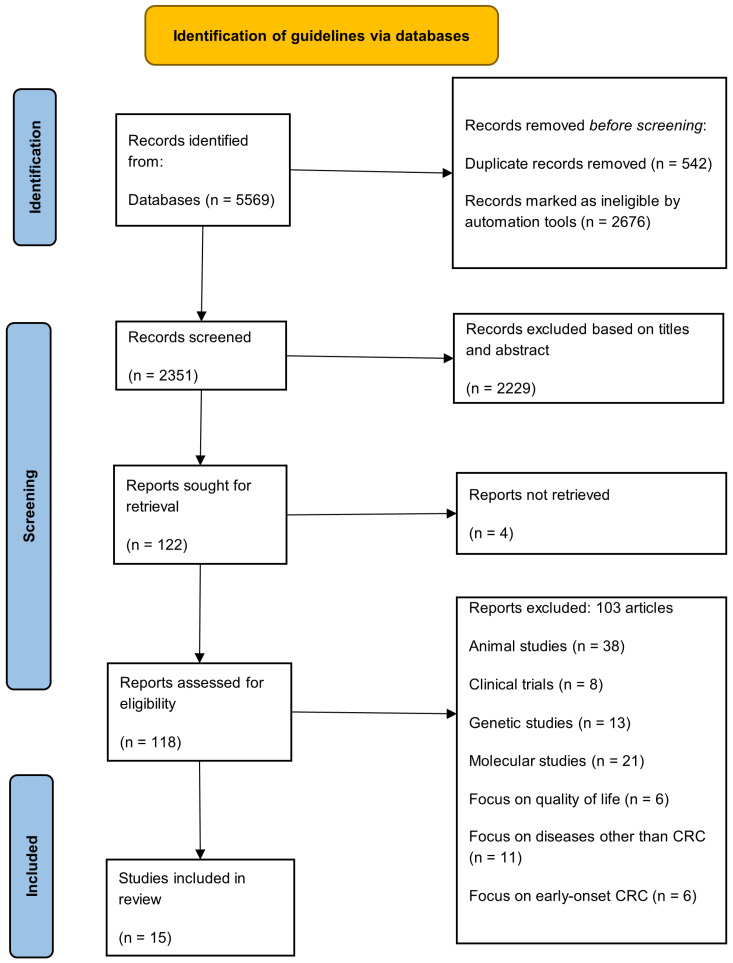
PRISMA flowchart.

**Figure 2 nutrients-14-01555-f002:**
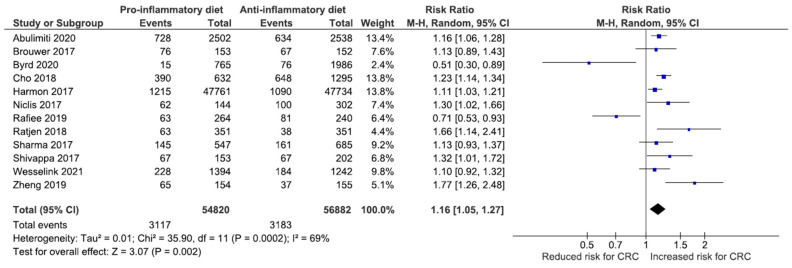
Forest plot for 12 studies comparing the risk of CRC between high (pro-inflammatory) and low (anti-inflammatory) scores.

**Figure 3 nutrients-14-01555-f003:**
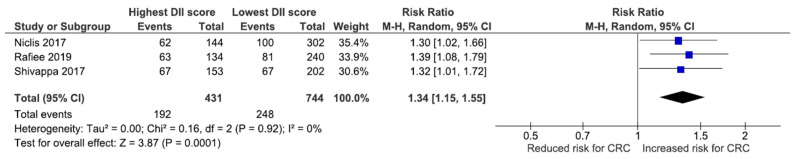
Forest plot for three studies relating the risk of CRC to continuous DII scores.

**Table 1 nutrients-14-01555-t001:** Keyword search used in the identification process.

Database	Search String
*Scopus*	TITLE-ABS-KEY ((“dietary inflammatory index” OR “dietary inflammatory score” OR “diet-related inflammation” OR “dietary inflammatory potential” OR “proinflammatory diet” OR “anti-inflammatory diet”) AND (“colorectal cancer *” OR “colorectal neoplas *” OR “colorectal tumo * r” OR “colorectal malignanc *”))
*Web of Science*	TS = ((“dietary inflammatory index” OR “dietary inflammatory score” OR “diet-related inflammation” OR “dietary inflammatory potential” OR “proinflammatory diet” OR “anti-inflammatory diet”) AND (“colorectal cancer *” OR “colorectal neoplas *” OR “colorectal tumo * r” OR “colorectal malignanc *”))
*PubMed*	((“dietary inflammatory index” OR “dietary inflammatory score” OR “diet-related inflammation” OR “dietary inflammatory potential” OR “proinflammatory diet” OR “anti-inflammatory diet”) AND (“colorectal cancer” OR “colorectal neoplasm” OR “colorectal tumor” OR “colorectal tumour” OR “colorectal malignancy” OR “colorectal malignancies”))
*EBSCOHost*	((“dietary inflammatory index” OR “dietary inflammatory score” OR “diet-related inflammation” OR “dietary inflammatory potential” OR “proinflammatory diet” OR “anti-inflammatory diet”) AND (“colorectal cancer” OR “colorectal neoplasm” OR “colorectal tumor” OR “colorectal tumour” OR “colorectal malignancy” OR “colorectal malignancies”))

The symbol * is used in the search strategy as truncation and wildcard function for keywords variation purposes.

**Table 2 nutrients-14-01555-t002:** Quality appraisal of selected studies using Newcastle–Ottawa quality assessment scale for case control and cohort studies.

**Studies (Case-Control Studies)**	**Selection (Maximum ****)**				**Comparability (Maximum **)**	**Exposure (Maximum ***)**			**Total Scores (Maximum 9)**
	**Is the Case Definition Adequate?**	**Representativeness of the Cases**	**Selection of Controls**	**Definition of Controls**	**Comparability of Cases and Controls on the Basis of the Design or Analysis**	**Ascertainment of Exposure**	**Same Method of Ascertainment for Cases and Controls**	**Non-Response Rate**	
Abulimi et al., 2020 [21]	*	*	*	*	**	-	*	-	7
Byrd et al., 2020 [22]	*	*	*	*	**	*	*	-	8
Cho et al., 2019 [23]	-	*	-	*	**	*	*	-	6
Niclis et al., 2018 [14]	*	-	*	*	*	-	-	*	5
Obon-Santacana, 2019 [24]	-	-	-	*	**	-	*	-	4
Rafiee et al., 2019 [25]	*	*	*	*	*	*	*	-	7
Sharma et al., 2017 [26]	-	*	*	*	*	*	*	-	6
Shivappa et al., 2017 [27]	*	*	-	*	*	*	*	-	6
Yuan et al., 2021 [28]	*	*	-	*	*	*	*	-	6
**Studies (cohort studies)**	**Selection (maximum ****)**				**Comparability (maximum **)**	**Outcome (maximum ***)**			**Total scores (maximum 9)**
	**Representativeness of the exposed cohort**	**Selection of the non-exposed cohort**	**Ascertainment of exposure**	**Demonstration that outcome of interest was not present at start of study**	**Comparability of cohorts on the basis of the design or analysis**	**Assessment of outcome**	**Was follow-up long enough for outcomes to occur**	**Adequacy of follow up of cohorts**	
Brouwer et al., 2017 [29]	*	-	*	*	*	*	*	*	7
Harmon et al., 2017 [30]	*	-	-	*	-	*	*	-	4
Ratjen et al., 2019 [31]	*	-	*	*	-	*	*	*	6
Tabung et al., 2017 [32]	*	*	*	*	*	*	*	*	8
Zheng et al., 2020 [33]	*	-	*	*	*	*	*	*	7
Wesselink et al., 2021 [34]	*	-	*	*	*	*	*	*	7

* denotes the scoring of one star; ** represents the scoring of two stars that is also the maximum scoring for the comparability domain; *** represents the scoring of three stars and the maximum scoring for the outcome domain; **** represents the scoring of four stars and the maximum scoring for selection domain.

**Table 3 nutrients-14-01555-t003:** Data extracted from the studies included for meta-analysis.

Author, Year	Study Location	Study Design	Study Period	Study Instrument	Number of Food Parameters	Sample Size	Range of DII Scores	Type of Data and Comparison	Measures of Association	Adjustment Factors
Abulimiti et al., 2020 [21]	China	Case control	2010–2019	81-item FFQ ^1^	34	2502 cases	−5.96 to +6.01	Categorical	OR = 1.40 (95% CI 1.16, 1.68)	Age, sex, marital status, residence, education level, occupation, income, BMI ^2^, smoking status, family history of CRC, comorbidities
						2538 controls		Quartile 4 vs. Quartile 1		
Brouwer et al., 2017 [29]	Netherlands	Prospective cohort	2006–2012	183-item FFQ	28	457	−11.7 to +8.4	Categorical	HR = 1.37 (95% CI 0.80, 2.34; *p* > 0.05)	Age, smoking status, education level
								Tertile 3 (0.3 to 8.4) vs. Tertile 1 (−11.7 to <−1.8)		
Byrd et al., 2020 [22]	United States	Case control	1991–2002	126-item FFQ	19	765 cases	(controls): −0.7 ± 2.4	Categorical	OR = 1.31 (95% CI 0.98, 1.75)	Age, sex, education, NSAIDs ^3^ use, hormone use, family history of CRC, smoking status, BMI, alcohol intake, physical activity
						1986 controls	(cases): −0.5 ± 2.4	Quintile 5 vs. Quintile 1		
Cho et al., 2019 [23]	Korea	Case control	2010–2013	106-item FFQ	35	632 cases	(controls): 0.94 ± 2.24	Categorical	OR = 1.38 (95% CI 1.12, 1.71)	Age, sex, family history of CRC, education level, BMI, physical activity, smoking status, alcohol intake
						1295 controls	(cases): 1.77 ± 1.97	High vs. Low		
Harmon et al., 2017 [30]	United States	Prospective cohort	1993–2010	169-item FFQ	28	190,963	−6.64 to +4.95	Categorical	HR = 1.21 (95% CI 1.11, 1.32)	Age, sex, race, comorbidities, smoking status, BMI, family history of CRC, education level, aspirin use, hormones use
								Quartile 4 (−0.52 to 4.95) vs. Quartile 1 (−6.64 to −3.66)		
Niclis et al., 2018 [14]	Argentina	Case control	2008–2015	127-item FFQ	22	144 cases	−3.15 to +3.77	Categorical	OR = 1.56 (95% CI 1.20, 2.03)	Age, sex, BMI, smoking status, socioeconomic status, physical activity, NSAIDs use
						302 controls		Tertile II (0.6–1.86) vs. Tertile 1 (<0.65)		
Obon-Santacana et al., 2019 [24]	Spain	Case control	2008–2013	140-item FFQ	30	1852 cases	(men): −5.11 to 5.47	Continuous DII (per one unit increase)	OR = 1.14 (95% CI 1.10, 1.18)	Sex, age, education level, study area, family history of CRC, smoking status, physical activity, BMI, NSAIDs use
						3447 controls	(women): −5.64 to 5.12			
Rafiee et al., 2019 [25]	Iran	Case control	2017–2018	148-items FFQ	21	134 cases	−4.23 to +3.89	Categorical	OR = 2.64 (95% CI 1.40, 4.99)	Age, sex, physical activity, salt intake, comorbidities, smoking, family history of CRC, cooking method, supplement intake
						240 controls		Tertile 3 (>0.04) vs. Tertile 1 (<−1.13)		
Ratjen et al., 2019 [31]	Germany	Prospective cohort	2009–2011	112-item FFQ	27	1404	−3.99 to +4.11	Continuous DII (per one unit increase)	HR = 1.08 (95% CI 0.97, 1.20)	Sex, age at diet assessment, BMI, physical activity, survival time, tumor location, metastasis, other type of cancers, therapy, smoking status, alcohol intake
Sharma et al., 2017 [26]	Canada	Case control	1999–2003	169-item FFQ	29	547 cases	−5.19 to +6.93	Categorical	OR = 1.65 (95% CI 1.13, 2.42)	Age, sex, BMI, physical activity, comorbidities, family history of CRC, smoking status, alcohol intake, NSAIDs use
						685 controls		Quartile 4 (≥0.3582) vs. Quartile 1 (<−2.036)		
Wesselink et al., 2021 [34]	Netherlands	Prospective cohort	2010–2017	204-item FFQ	28	1478	−12.2 to +8.5	Categorical	HR = 0.98 (95% CI 0.94, 1.04; *p* > 0.05)	Age, sex, staging, BMI, smoking status, NSAIDs use, comorbidities
								Tertile 3 (1.2 to <8.5) vs. Tertile 1 (−12.2 to <−1.0)		
Shivappa et al., 2017 [27]	Jordan	Case control	2010–2012	90-item FFQ	18	153 cases	−2.25 to +2.86	Continuous DII (per one unit increase)	OR = 1.45 (95% CI 1.13, 1.85)	Age, sex, education level, physical activity, BMI, smoking status, family history of CRC
						202 controls				
Tabung et al., 2017 [32]	United States	Prospective cohort	1993–2014	122-item FFQ	32	87,042	−6.62 to +5.39	Categorical	HR = 1.06 (95% CI 0.90, 1.26)	Age, race, education level, smoking status, comorbidities, regular NSAIDs use, estrogen use, BMI, physical activity
								Quintiles 5 vs. Quintiles 1		
Yuan et al., 2021 [28]	United States	Case control	2005–2015	175-item FFQ	34	587 cases	−5.9 to +4.6	Continuous DII (per one unit increase)	OR = 1.07 (95% CI 0.97, 1.19)	Age, gender, race, BMI, education level, smoking status, comorbidities, NSAIDs use, family history of CRC, supplements use
						1313 controls				
Zheng et al., 2020 [33]	United States	Prospective cohort	1993–2015	122-item FQ	32	161,808	−6.80 to +3.25	Categorical	HR = 0.72 (95% CI 0.46, 1.12)	Age, race, smoking status, income levels, cancer staging, education level, physical activity, BMI
								Tertile 1		
								(−5.96 to −2.25) vs. Tertile 3 (−0.18 to 3.82)		

^1^ FFQ, food frequency questionnaire; ^2^ BMI, body mass index; NSAIDs ^3^, non-steroidal anti-inflammatory drugs.

**Table 4 nutrients-14-01555-t004:** Subgroup analyses of studies reporting the risk for CRC between high (pro-inflammatory) and low (anti-inflammatory) scores.

Subgroups	No. of Studies	RR (95% CI)	Heterogeneity		Significance Test	
			I^2^ (%)	*p*	Z	*p*
Study design						
Case-control	7	1.14 (0.89, 1.45)	81%	0.000	1.03	0.300
Cohort	4	1.24 (1.06, 1.44)	63%	0.030	2.74	0.006
Groups						
Continuous	4	0.35 (0.28, 0.41)	0%	0.400	10.12	0.000
Categorical	3	1.61 (1.26, 2.05)	0%	0.900	3.80	0.000
Region						
AMR	4	0.32 (0.24, 0,40)	62%	0.050	8.29	0.000
EUR	4	0.40 (0.33, 0.47)	98%	0.000	10.50	0.000
Asia	2	0.44 (0.34, 0.54)	95%	0.000	8.59	0.000
EMR	2	0.36(0.21, 0.52)	22%	0.260	4.61	0.000
Study period						
Less than 10 years	11	1.12 (0.94, 1.35)	97%	0.000	1.27	0.200
10 years or more	2	2.95 (2.47, 3.52)	92%	0.001	12.01	0.000
Adjustment for family history of CRC						
Yes	8	1.01 (0.82, 1.24)	97%	0.000	0.06	0.950
No	5	1.31 (1.10, 1.56)	55%	0.060	3.01	0.003
Adjustment for education level						
Yes	8	1.11 (0.89, 1.39)	98%	0.000	0.93	0.350
No	5	1.12 (0.90, 1.39)	75%	0.003	1.04	0.300
Adjustment for comorbidities						
Yes	5	1.08 (0.97, 1.20)	64%	0.030	1.41	0.160
No	8	1.18 (0.92, 1.50)	96%	0.000	1.28	0.200
Adjustment for physical activity						
Yes	9	1.11 (0.89, 1.39)	95%	0.000	0.93	0.350
No	4	1.13 (1.07,1.20)	0%	0.890	4.38	0.000
Adjustment for BMI						
Yes	5	1.60 (1.54, 1.67)	96%	0.000	23.81	0.000
No	2	0.86 (0.78, 0.96)	92%	0.001	2.84	0.004

## Data Availability

Not applicable.

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
