# Peer review of "Meta-Analysis of the Association between Dietary Inflammatory Index (DII) and Colorectal Cancer"

_nutrients, 2022, doi:10.3390/nu14081555_

Round 1

Reviewer 1 Report

This is a good study linking the association between "Dietary Inflammatory Index (DII) and Colorectal Cancer".
The authors linked a very likely cause, i.e. diet type to the risk of CRC.
My remarks that can contribute to improving the quality of work are:
- Please state in the Methods section why only studies published between 2017 and 2021 were included in the analysis.
Although the authors state in the Introduction that the studies published in previous years (Fowler 2017) dealt with the western type diet, in addition to studies from Europe and America, they also included studies from Australia, Korea, Jamaica, and Iran, so it cannot be said that only the western diet type was analyzed. The same variables were also examined and the same results were obtained.
- Another major objection is that both the variables examined, diet and sedentary lifestyle ultimately lead to an increase in BMI. BMI has been researched in at least 13 of these studies and presents a result that will interest readers of this article. The authors did not mention its significance in a single sentence. This addition would represent a good new contribution to the significance of the study, even in the case of a negative finding, because in an earlier meta-analysis (Fowler 2017) that already had the same results and population, BMI data were not analyzed.
If this changes will not done the study is a simple replica of a paper published by Mackenzie E. Fowler, Tomi F. Akinyemi from 2017.

Author Response

1. Please state in the Methods section why only studies published between 2017 and 2021 were included in the analysis.

Author response:

Despite the consistency of DII as an instrument in substantial evidence-based research, the additional food parameters and revised scoring algorithms enhanced the comparability between studies [19]. Considering the utilization of revised version DII in recently published studies concerning colorectal cancer justifies the selection of the latest five years duration. 

The author had made corrections in line 102-106.

2. Although the authors state in the Introduction that the studies published in previous years (Fowler 2017) dealt with the western type diet, in addition to studies from Europe and America, they also included studies from Australia, Korea, Jamaica, and Iran, so it cannot be said that only the western diet type was analyzed. The same variables were also examined and the same results were obtained.

Author response:

To date, the evidence on the usefulness of the DII in predicting the risk of CRC is established mainly focus on the western diet with little input from other parts of the world based on the studies published before 2017 [14,15].

The author had made corrections in line 59-62

3. Another major objection is that both the variables examined, diet and sedentary lifestyle ultimately lead to an increase in BMI. BMI has been researched in at least 13 of these studies and presents a result that will interest readers of this article. The authors did not mention its significance in a single sentence. This addition would represent a good new contribution to the significance of the study, even in the case of a negative finding, because in an earlier meta-analysis (Fowler 2017) that already had the same results and population, BMI data were not analyzed.
If this changes will not done the study is a simple replica of a paper published by Mackenzie E. Fowler, Tomi F. Akinyemi from 2017.

Author response:

The interplay of modifiable risk factors such as unhealthy dietary patterns and sedentary lifestyles have partly contributed to the complexity of obesity to an extent [49]. Several epidemiological evidence have demonstrated that individuals with higher-than-normal BMI are likely to have a higher risk for CRC [49,50,51,52,53]. In a large population-based cohort trial, obesity during early adulthood and the constantly increasing BMI throughout the lifespan were significantly associated with CRC [54]. Recent genetic studies suggested that for every one-unit increase in genetically predicted BMI, there is an increase in the odds ratios for CRC [55,56,57], indirectly implies the causality relationship between obesity and CRC. Although the current review reported statistically significant heterogeneity between studies with adjustment for BMI, future studies should consider the influence of residual confounding to delineate the true effect of BMI on CRC.

The authors had made corrections in line 282-292

Reviewer 2 Report

The Authors presented a well-written meta-analysis of the association between dietary inflammatory index and colorectal cancer. 

Even if the argument is per se well known, the Authors established a strong correlation between dietary intakes and the RR. of developing colorectal cancer.

The review is well structured and written.

The figures are not immediately readable, please check the size and/or the pixels/inch, because they look blurry.

Author Response

  1. The figures are not immediately readable, please check the size and/or the pixels/inch, because they look blurry.

Author response:

The author has improved the figures in Figure 2 and Figure 3
